# Asymmetry Optimization for 10 THz OPC Transmission over the C + L Bands Using Distributed Raman Amplification

**DOI:** 10.3390/s23062906

**Published:** 2023-03-07

**Authors:** Paweł Rosa, Giuseppe Rizzelli Martella, Juan Diego Ania Castañón, Mingming Tan

**Affiliations:** 1National Institute of Telecommunications, Szachowa 1, 04-894 Warsaw, Poland; 2LINKS Foundation, Via Piercarlo Boggio 61, 10138 Torino, Italy; 3Instituto de Óptica, IO-CSIC, Serrano 121, 28006 Madrid, Spain; 4Aston Institute of Photonics Technologies, Aston University, Birmingham B4 7ET, UK

**Keywords:** Raman amplification, optical fiber communications, optical phase conjugation

## Abstract

An optimized design for a broadband Raman optical amplifier in standard single-mode fiber covering the C and L bands is presented, to be used in combination with wideband optical phase conjugation (OPC) nonlinearity compensation. The use of two Raman pumps and fiber Bragg grating reflectors at different wavelengths for the transmitted (C band) and conjugated (L band) WDM channels is proposed to extend bandwidth beyond the limits imposed by single-wavelength pumping, for a total 10 THz. Optimization of pump and reflector wavelength, as well as pump powers, allows us to achieve low asymmetry across the whole transmission band for optimal nonlinearity compensation. System performance is simulated to estimate OSNR, gain flatness and nonlinear Kerr distortion.

## 1. Introduction

Multiple solutions have been proposed over the past decades to address the critical problem posed by Kerr nonlinearity’s cap to capacity in optical fiber communications [1]. Although digital compensation techniques have been successfully applied to the mitigation of nonlinear effects, they are inextricably associated to an increase in computational cost and energy consumption, and thus the possibility of finding a solution to the problem that works on the physical layer, is a very attractive one. Using optical phase conjugation (OPC) in the middle of the optical fiber link is a particularly effective way to combat the nonlinearities [1,2,3,4,5,6,7,8,9,10,11,12,13,14], which allowed for the first demonstration of optical communications above Shannon’s limit [10]. The technique, is, however, not free from technical challenges in terms of implementation. To maximize its efficiency when applying it to multi-channel nonlinearity compensation, there are a few approaches which will need to be implemented on the fiber link. For example, in [15,16], the fiber nonlinearity compensation using a mid-link OPC can be achieved using a symmetrical chromatic dispersion slope or effective management of dispersion mapping before and after the OPC. Alternatively, in [17,18,19,20,21,22,23,24], with a purposefully designed distributed Raman amplification scheme, a symmetrical signal power profile along the fiber before and after the OPC was demonstrated to maximize the effectiveness of nonlinearity compensation with a mid-link OPC. In [24], we numerically optimised the in-span signal power asymmetry for three different advanced Raman amplification schemes using a single channel in the middle of the C-band at 1545 nm and identified that the second-order distributed Raman amplification based on a single-side FBG random distributed feedback laser is the most convenient design to achieve the best signal power profile symmetry. Next, in [21] we advanced our simulations to demonstrate the possibility of WDM transmission across the C band, by optimising symmetry over a broad section (40 channels with a 25 GHz spacing). Recently, ultra-wideband or multi-band optical transmission assisted by Raman amplification has been a hot topic of discussion [25,26,27,28], as an efficient tool to fully unlock the potential transmission capacity of single mode fiber (SMF). In this context, the possibility of applying the second-order Raman amplification scheme discussed in [17,29,30,31] in other transmission bands, using a fiber Bragg grating in another band with appropriate Raman pump wavelengths, becomes particularly interesting. In particular, in [17] we showed a method for bandwidth extension using a fixed wavelength Raman pump centered at 1366 nm, while using FBGs at different wavelengths for the originally transmitted and the conjugated channels. Utilizing this method, we managed transmission over 6 THz with a 5.9% average asymmetry in a 192–198 THz band using 60 km SMF span. In [18] we summarized our experimental work reviewing several configurations of distributed Raman amplifiers designed specifically for fiber nonlinearity compensation in a mid-link optical phase conjugation system, demonstrating that, for nearly symmetrical signal power profiles, the Raman schemes in both the single-span and two-span systems provide a 9 dB enhancement of the nonlinear threshold in a 200 Gb/s DP-16QAM transmission system using a mid-link OPC

In this paper, we make use of a random distributed feedback fiber Raman laser amplifier scheme and we significantly extend the working bandwidth to a full C band (before the OPC) and L band (after the OPC) optimizing the amplifier specifically for dual-band OPC with a total bandwidth of 10 THz. Unlike in previous theoretical and experimental work, where we used a fixed single Raman pump wavelength, in this case we consider the wavelengths and optical power of the primary Raman pumps and the wavelengths of the secondary FBG mirrors as optimisable parameters to find the best average symmetry in multi-channel C and L band transmission systems with mid-link OPC. 

## 2. Amplifier Design for Optical Phase Conjugation

In the recent past, the design of optical networks was based exclusively on lumped amplification and constrained to the C band due to the convenience, reliability and cost effectiveness of Erbium-doped fiber amplifiers (EDFAs). However, the exponential increase of data traffic over the internet has pushed further bandwidth extensions towards the optical L and S bands, which are not easily achievable with doped fibers alone. In this context, Raman amplification, whether lumped or distributed, offers an attractive alternative. 

Higher order Raman amplification is well known for improving transmission performance through improved noise performance, achieving extended bandwidth even with a single pump wavelength [17]. At the same time, distributed amplification allows for a precise control of signal power variation across transmission fibre [24], which is necessary to minimize asymmetry for the OPC system. In our previous works [17,18,19,20,21,22,23,24] we compared several designs of distributed Raman amplifiers: first order, second order and dual order using bi-directional and backward-only pumping schemes. Bi-directionally pumped distributed Raman amplification with a single FBG at the end of the transmission span performed best in terms of asymmetry [24] and relative intensity noise (RIN) [30,31,32], which is a key design feature for data transmission in a 60 km span, hence we continue with this design and modifications to meet our bandwidth requirement.

In order for nonlinear impairments to be perfectly compensated in a system with mid-link optical phase conjugation, the following ideal condition must be fulfilled for each of the channels:(1)$$\;\;\;\;\;\;\;\;\;\;\;\;\;\;\;\;\;\;\;\;\;\;\frac{\beta_{2}\left(\;L_{OPC}-z\right)}{\gamma \left(L_{OPC}-z\right)P\;\left(L_{OPC}-z\right)}=\frac{\beta_{2}^{’}\left(L_{OPC}+z\right)}{\gamma^{\prime }\left(L_{OPC}+z\right)P^{\prime }\left(L_{OPC}+z\right)}$$
where $\beta_{2}$ represents the dispersion coefficient for the channel wavelength, $\beta_{2}^{’}$ is its equivalent for the conjugate channel, γ and $\gamma^{\prime }$ represent the nonlinear coefficients for the original and conjugate channels and *P* and *P’* indicate the corresponding signal powers. *L_OPC_* indicates the position of the optical phase conjugator, and *z* ranges from *0* to *2L_OPC_*. The key to maximize performance in OPC-assisted systems lies in reducing signal power asymmetry between P and $P^{\prime }$.

Dispersion and nonlinearity coefficients at the wavelengths of the original and conjugated channels depend on fiber characteristics, and can be very similar in modern commercially available SMFs, so optimization options are limited to signal power evolution, which must be made as symmetrical as possible before and after the mid-link OPC for the original and conjugate channels. In practice, and since long-haul communications rely on the use of periodic amplification cells, the more efficient approach [21,24], is to aim for symmetric power evolution with respect to the periodic span mid-point, as well as similar power variation levels on both the original and conjugate channels, defining an asymmetry parameter to be optimized (see Section 3, below). 

In our research to design an amplifier spanning a 10 THz bandwidth we independently simulated schemes based on different wavelengths for the first and second order pumps (i.e., the FBG center wavelengths) with various forward and backward pump powers for a transmitted (5 THz C band) and conjugated (5 THz L band) wavelength division multiplexed (WDM) grid with a 100 GHz spacing. 

The schematic design of an amplifier for the OPC-based transmission system is shown in Figure 1. Primary forward and backward Raman pump frequencies $v_{p_{1}}$, as well as the central frequency of the FBG, $v_{p_{2}}$ were chosen accordingly to the target amplification bandwidth, aiming for the best asymmetry performance in a 60 km span length. Forward pump powers $P_{p_{1}}^{+}\;$of the first order Raman laser for both bands were simulated from 0.7 to 1.4 W with a 100 mW step. Backward pump powers$\;P_{p_{1}}^{-}\;$were simulated to give 0 dB net gain for a channel under test, and then all remaining WDM channels were simulated with fixed pump powers. The FBG (200 GHz bandwidth) located at the end of the transmission line reflects backscattered Rayleigh Stokes-shifted light from the backward pump$\;P_{p_{1}}^{-}\;$and form a random DFB laser acting as secondary backward pump $P_{p_{2}}^{-}$ that amplifies the WDM signal at the C or L transmission band. The transmitted power per channel is set to −10 dBm.

### 2.1. C Band: 50 Transmitted Channels with 100 GHz Spacing 191.2–196.1 THz

To study the performance of the amplifier in the 5 THz C band (1528.77–1567.95 nm) the wavelength of the first order pump is made to range from 1362 to 1374 nm, whereas the wavelength of the FBG ranges from 1456 to 1474 nm.

### 2.2. L Band: 50 Conjugated Channels with 100 GHz Spacing 186.2–191.1 THz

To study the performance in the 5 THz L band (1568.77–1610.06 nm) we simulated for wavelengths of the first order pump ranging from 1402 to 1414 nm, and wavelength of the FBG ranging from 1492 to 1508 nm.

## 3. Simulation Parameters 

To simulate our 10 THz wide band WDM OPC system we used our model of a second order Raman amplifier with a single FBG mirror at the end of the transmission span, that was derived and developed from [33]. The transmission band (C or L) was amplified by the gain from the primary Raman pump in forward $P_{p_{1}}^{+}$ and backward $P_{p_{1}}^{-}$ directions as well as secondary pump in the backward direction $P_{p_{2}}^{-}$ generated at the wavelength of the FBG reflector.
$$\frac{dP_{p1}^{\pm }}{dz}=\mp \left(\alpha_{p_{1}}P_{p_{1}}^{\pm }\right)\mp \frac{g_{p_{1}\to p_{2}}}{A_{eff}}\frac{v_{p_{1}}}{v_{p_{2}}}\left(P_{p_{2}}^{+}+P_{p_{2}}^{-}+4hv_{p_{2}}\Delta v_{p_{2}}\left(1+\frac{1}{e^{\frac{h(v_{p_{1}}-v_{p_{2}})}{K_{B}T}}-1}\right)\right)P_{p_{1}}^{\pm }\mp$$
(2)$$\frac{g_{p_{1}\to s}}{A_{eff}}\frac{v_{p_{1}}}{v_{s}}\left(P_{s}+n_{s}^{+}+n_{s}^{-}+4hv_{s}\Delta v_{s}\left(1+\frac{1}{e^{\frac{h(v_{p_{1}}-v_{s})}{K_{B}T}}-1}\right)\right)P_{p_{1}}^{\pm }\mp \epsilon_{p_{1}}P_{p_{1}}^{\pm }$$
$$\frac{dP_{p2}^{-}}{dz}=\left(\alpha_{p_{2}}P_{p_{2}}^{-}\right)-\frac{g_{p_{1}\to p_{2}}}{A_{eff}}\left(P_{p_{2}}^{-}+2hv_{p_{2}}\Delta v_{p_{2}}\left(1+\frac{1}{e^{\frac{h(v_{p_{1}}-v_{p_{2}})}{K_{B}T}}-1}\right)\right)\left(P_{p_{1}}^{+}+P_{p_{1}}^{-}\right)+$$
(3)$$\frac{g_{p_{2}\to s}}{A_{eff}}\frac{v_{p_{2}}}{v_{s}}\left(P_{s}+n_{s}^{+}+n_{s}^{-}+4hv_{s}\Delta v_{s}\left(1+\frac{1}{e^{\frac{h(v_{p_{2}}-v_{s})}{K_{B}T}}-1}\right)\right)P_{p_{2}}^{-}-\epsilon_{p_{2}}$$

where $P_{p}^{\pm }$ are the powers of the forward (+) or backward (-) propagating pump, α is the corresponding attenuation, $A_{eff}$ is the effective core area, g is the Raman gain coefficient depending on the frequency shift of the lasing and each WDM signal’s wavelength for a standard single mode fiber as in Figure 2. 

$n_{s}^{+}\;\mathrm{and}\;n_{s}^{-}$ are the forward and backward noise at the frequency of the signal, v is the frequency and Δv is bandwidth of each component: $p_{1}$ (primary pump), $p_{2}$ (secondary pump) and *s* (signal). h is the Planck’s constant, $K_{B}$ is the Boltzmann constant and *T* is the absolute temperature. ϵ is the Rayleigh backscattering coefficient.
(4)$$\frac{dP_{D}}{dz}=-\left(\;\alpha_{D}P_{D}\right)+\frac{g_{p_{2}\to D}}{A_{eff}}\left(P_{p_{2}}^{-}\right)\left(\;P_{D}+2hv_{D}\Delta v_{s}\left(1+\frac{1}{e^{\frac{h(v_{p_{2}}-v_{D})}{K_{B}T}}-1}\right)\right)+\;\frac{g_{p_{1}\to D}}{A_{eff}}\left(P_{p_{1}}^{+}+P_{p_{1}}^{-}\right)\left(\;P_{D}+2hv_{D}\Delta v_{s}\left(1+\frac{1}{e^{\frac{h(v_{p_{1}}-v_{D})}{K_{B}T}}-1}\right)\right)+\epsilon_{s}n_{D}^{-}\;\;\;$$

Our model also takes into account the accumulated light power $P_{D}\;$from all WDM channels and both pumps that is depleting amplification gain (it is then added or subtracted to Equations (2) and (3) in our simulations), double Rayleigh scattering (DRS), and amplified spontaneous emission (ASE) noise (calculated in 0.1 nm bandwidth) from each spectral component in the transmission band. The values of the Rayleigh backscattering coefficients for primary pump $P_{p}^{\pm }$, lasing $P_{p_{2}}^{-}\;$and at the frequencies of the signal $v_{s}\;$channels are assumed to be 1.0 × 10^−4^, 6.5 × 10^−5^ and 4.5 × 10^−5^ km^−1^, respectively. The bandwidth of the FBG $P_{p_{2}}^{-}$ in the simulations was set to 200 GHz. With relatively low input power per channel (−10 dBm) and channel spacing of 100 GHz we do not consider cross-gain modulation in our simulations. The span length was 60 km. The asymmetry for each channel was calculated using the formula: (5)$$\;\;\;\;\;\;\;\;\;\;\;\;\;\;\;\;\;\;\;\;\;\;\;\;\;\;\;\;\frac{\int_{0}^{L}\left|P_{1}\left(z\right)-P_{2}\left(L-z\right)\right|dz}{\int_{0}^{L}[P_{1}\left(z\right)+P_{2}\left(z\right)]/2dz}\times 100$$
where *L* is the span length, *P*_1_ and *P*_2_ represents signal power evolution of the transmitted and conjugated channels, respectively. 

The coefficients in the simulations were adjusted to match our experimental measurements of the signal power variation (SPV) in the SMF span. To measure the SPV, a laser source at 1545 nm with launch power of 0 dBm was used to provide a probe signal whose power evolution along the 80, 100 and 120 km transmission span was then monitored using a standard OTDR [34]. Results of the OTDR traces (noisy) and simulations (solid) are shown in Figure 3.

To verify the accuracy of the simulations we also measured asymmetry using modified OTDR system [34] in a 60 km SMF span for various forward and backward pump power rations. The results of the simulations are shown in Figure 4 (red). There is a very good agreement between experimental measurements and numerical simulations. 

## 4. Results and Discussion

To evaluate the optimum configuration for the lowest asymmetry across whole transmission spectrum we verified the results obtained with each original pump wavelength against different FBG (for original pump 1) and conjugated pump wavelengths and FBGs (for conjugated pump 2). As an example, below in Figure 5 we show the optimization process for a primary pump centered at 1364 nm and FBG ranging from 1456–1462 nm. For clarity, the results for L band pump (pump 2) wavelength (1402–1414 nm) are already given for the optimum (best average asymmetry match) FBG (simulated from 1492–1508 nm).

The asymmetry difference between the worst and best performing channels shown in Figure 6 is heavily biased due to the first WDM channel in the C band (CH1) that is off the grid of the Raman amplification gain. This is explained and shown with the further results where we present signal power variation, asymmetry and on-off gain for each individual channel in a 10 THz band. 

The best primary pump wavelength offset between the transmitted and corresponding conjugated WDM grid was found to be 48 nm: for the primary pump in C band centered at 1364 nm, the best matching primary pump for the L band was 1412 nm. In Figure 7 we show the best average asymmetry performance for all 100 WDM channels (50CH in C band versus 50CH in L band) as a function of primary pump and optimized FBG wavelengths for the conjugated L band channels. The choice of the wavelength of the FBG was previously investigated and shown in Figure 5. In this case the best asymmetry performance gave FBG centered at 1458 nm, with an average asymmetry below 10%. 

Using the same methodology, we evaluated the primary pump wavelengths for transmitted C band channels ranging from 1362 to 1374 nm and for conjugated L band channels from 1402 to 1414 nm. Additionally, for each pump wavelength we simulated a range of different FBGs for the transmitted C band: 1456 to 1474 nm and conjugated L band channels 1492 to 1508 nm, with a 2 nm step for all cases, giving us proximately 1.5 × 10^6^ possible combinations (pump wavelength × FBG × pump power × possible channel optimizations $\left(50\times 50\right)$).

Out of all available combinations, the best performing configuration giving an average asymmetry of 8.2% across all WDM channels was achieved with the distributed Raman amplifier settings shown in Table 1 below.

The results of the best performing configuration with primary C band pump at 1370 nm as a function of primary L band pump for conjugated channels with optimized FBG are shown in Figure 8 (red). For reference we also show the discussed results for the primary C band pump wavelength centered at the 1364 nm (blue). We can notice that for a 6 nm (1364 to 1370 nm) shift in primary pump wavelength for the C band, the choice of the best matching L band primary pump wavelength would only change by 2 nm from 1412 nm (blue) to 1410 nm (red). However, we would like to stress that the average asymmetry is highly biased by the few worst performing channels, with performances that are off by 20–30%, while the rest varies by +/− 2%, hence the choice of optimal wavelengths of the primary pumps and FBGs is not simple, and will depend on system needs and circumstances. This issue becomes even more evident if we start leveraging the negative impact of RIN on actual data transmission due to high forward pump powers and the benefit achieved from lower averaged overall asymmetry performance. In [31], the authors show that lower signal power variation due to higher forward pumping does not necessarily translate to better actual transmission performance. This problem can be mitigated using broadband forward pump power [31], which justifies our choice of higher order Raman amplification without direct forward lasing, one Stokes down-shifted from the band of the amplified signal. 

The impact of the forward pump power on the asymmetry of each WDM channel in the OPC system is shown in Figure 9. We can notice that the asymmetries of the first seven channels are practically immune to forward pumping power and do not vary significantly. This can be explained with Figure 10, where we the signal power variations (SPV) for each individual channel in C and L band are displayed. Higher SPV and asymmetry mismatch are directly related to the gain performance of our amplifier. In Figure 11 we show the best possible over all on-off gain for all channels (blue) as well gain performance for each channel at the best asymmetry performance configuration (red) in 10 THz C + L band distributed Raman amplification. The best gain flatness, with about 3 dB gain variation, was achieved for the configuration with the primary C band pump centered at 1370 nm with the FBG at 1460 nm. The primary L band pump was set to 1408 nm with the FBG at 1498 nm. Gain performance at best asymmetry (blue) is shown for configuration as in Table 1. 

The best asymmetry performance between the transmitted and corresponding conjugated channel was channel 18, giving the lowest asymmetry of 2.82%. The power profiles of both channels are shown below in Figure 12. We may notice a very low signal power variation of 1.74 dB or less for the transmitted and conjugated channels across the whole 60 km raw distributed Raman transmission span. 

In Figure 13 we show the theoretical prediction of Four wave mixing (FWM) power comparison using a mid-link OPC configuration (red) and a raw transmission (blue). The FWM power (defined in ref [20,35]) in the best scenario was suppressed by over 45 dB in a low frequency range and 40 dB at its peak just below 20 GHz. That demonstrates that the nonlinear distortion limiting the capacity of long-haul optical communication systems can be efficiently controlled with the fine optimization of the mid-link OPC in a real time data transmission. FWM nonlinearity compensation may be also limited by using various techniques of digital signal processing, however, this solution is computationally expensive and time consuming which, at the current state of art of the computational power, does not really allow for an advanced real time transmission.

Finally, in Figure 14 we show the optical signal to noise ratio (OSNR) performance calculated over a 0.1 nm bandwidth as the difference between the signal power and the noise power as well as nonlinear phase shift (NPS) results for all transmitted and conjugated channels in an optimized 10 THz WDM grid (186.2–196.1 THz) in a 60 km standard single mode span. The OSNR varies from just below 39 dB to 40.5 dB, which is a very good performance across such a wide bandwidth with a raw Raman amplified transmission. NPS variation is also very low across all transmission bandwidth, with the lowest performance in front of C and L bands.

## 5. Conclusions

Using numerical simulations based on experimental results, we propose and demonstrate, for the first time, an amplifier design for C + L band mid-link OPC transmission achieving the lowest average asymmetry up to date over a 10 THz bandwidth. Using half-open cavity random DFB Raman laser amplification with two different pump wavelengths for the transmitted and corresponding conjugated channels in combination with different FBGs we successfully extend the operating bandwidth of the mid-link OPC setup, obtaining very promising performance results. The optimized system is capable of 10 THz transmission with OSNR values above 38.8 dB and an average asymmetry of 8.2% for all WDM channels. The best possible configuration shows gain flatness below 3 dB across the 10 THz grid in a raw Raman transmission without any gain flattening filters applied.

## Figures and Tables

**Figure 1 sensors-23-02906-f001:**
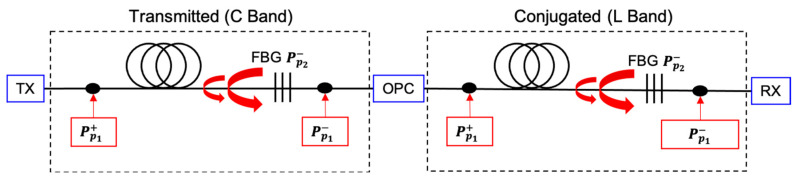
Raman fiber laser amplifier for C and L band with a half-open cavity random lasing designed for a 10 THz OPC-based transmission system.

**Figure 2 sensors-23-02906-f002:**
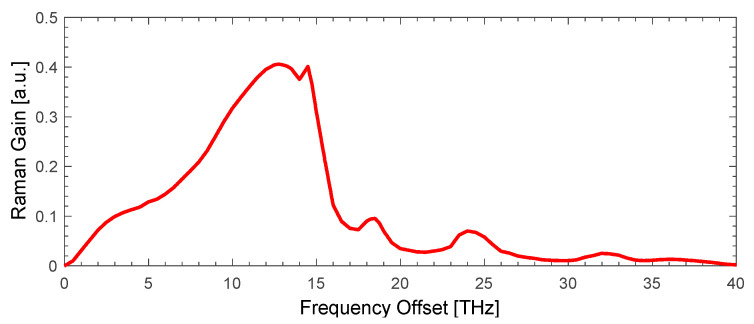
Normalized Raman gain spectrum in single mode silica fiber.

**Figure 3 sensors-23-02906-f003:**
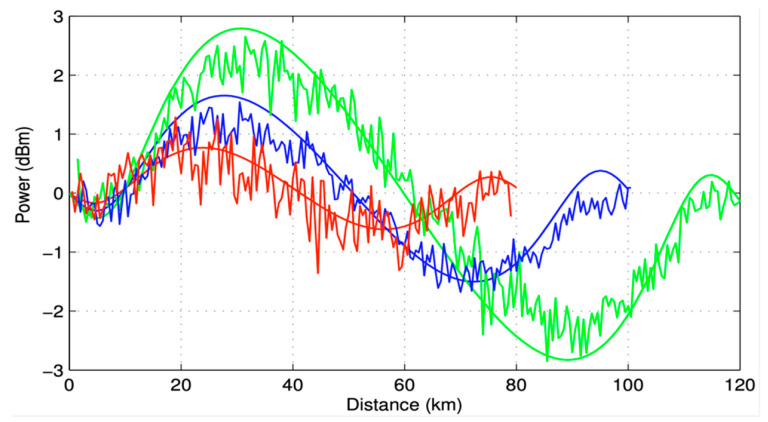
Experimental power evolution in 80 km (red), 100 km (blue) and 120 km (green) span taken using a modified OTDR system and a simulation fit in a SMF span.

**Figure 4 sensors-23-02906-f004:**
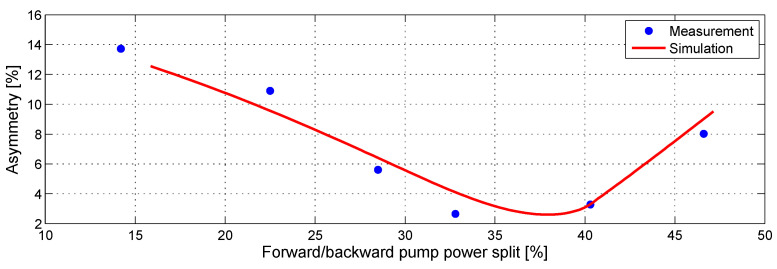
Asymmetry match in a 60 km SMF span for different forward and backward pump power split.

**Figure 5 sensors-23-02906-f005:**
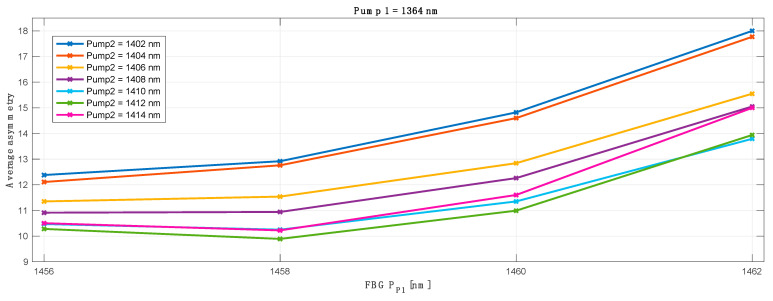
Primary pump and FBG wavelength optimization as a function of average asymmetry for all transmitted and conjugated channels. *X*-axis show FBG optimization for the primary C band pump fixed at 1364 nm.

**Figure 6 sensors-23-02906-f006:**
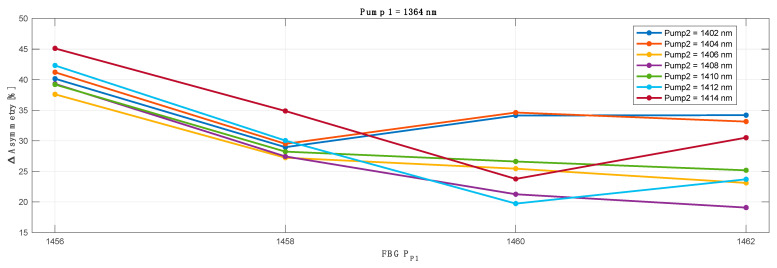
Asymmetry difference between worst and best performing channel in a 50 channels transmission band based on the results shown in Figure 5.

**Figure 7 sensors-23-02906-f007:**
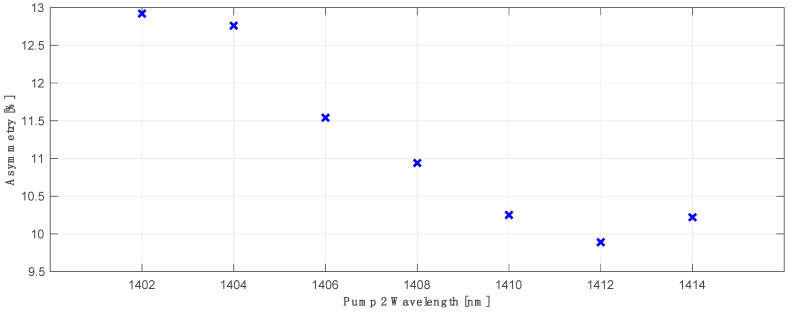
Optimum asymmetry optimization for primary C band pump at 1364 nm and FBG 1458 nm as a function of primary L band pump wavelength for conjugated channels with optimized FBG.

**Figure 8 sensors-23-02906-f008:**
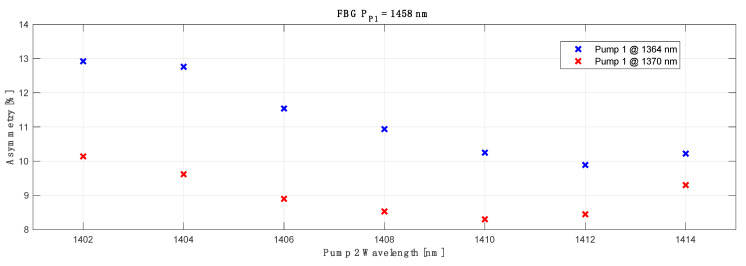
Best performing configuration with primary C band pump at 1370 nm (red) as a function of primary L band pump for conjugated channels with optimized FBG. The 1364 nm pump (blue) is plotted as the reference to previous investigation.

**Figure 9 sensors-23-02906-f009:**
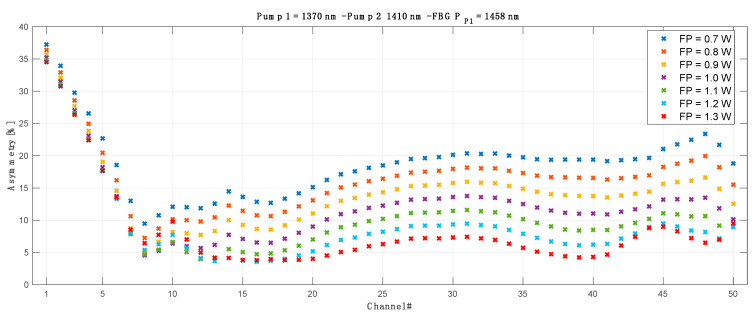
Impact of the forward pump power on asymmetry of each channel in best performing configuration as in Table 1.

**Figure 10 sensors-23-02906-f010:**
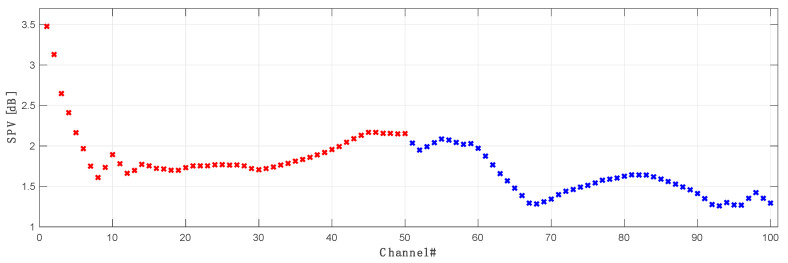
Signal power variation of all 100 channels in a 10 THz OPC band. Original C band channels are marked red, while L band is marked blue.

**Figure 11 sensors-23-02906-f011:**
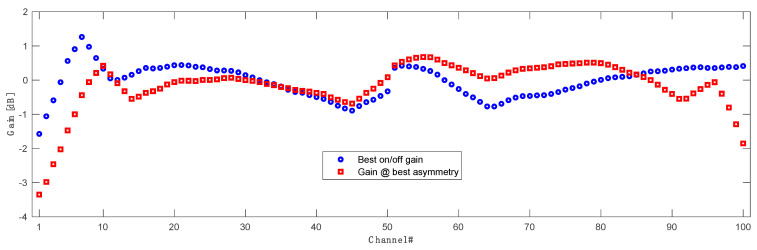
Gain performance of proposed Raman amplifier showing best possible on/off gain (blue) and actual gain (red) for best asymmetry of all 100 channels in a 10 THz transmission.

**Figure 12 sensors-23-02906-f012:**
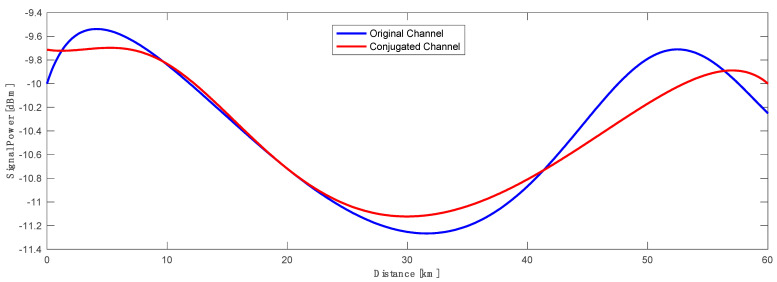
Signal power profile of original (blue) CH#18 and corresponding conjugated channel (red) with asymmetry of 2.82% in a 60 km standard SMF span.

**Figure 13 sensors-23-02906-f013:**
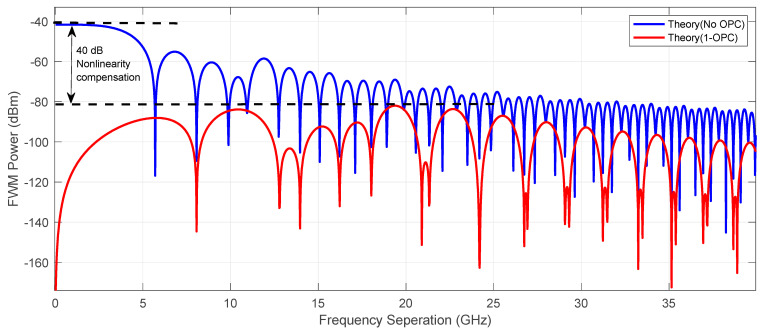
Theoretical prediction of Four wave mixing (FWM) power as a function of frequency separation for mid link OPC link (red) and without OPC (blue) for the best performing channel shown in Figure 12.

**Figure 14 sensors-23-02906-f014:**
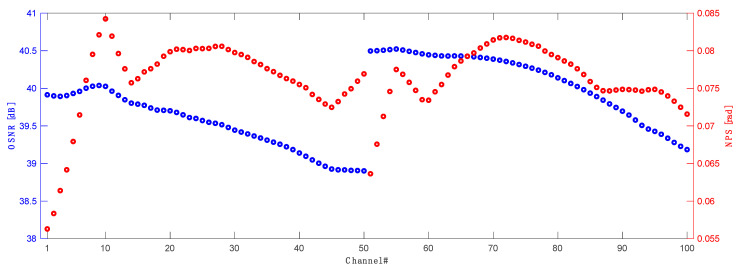
OSNR (blue) and NPS (red) performance across 10 THz bandwidth covering all C band (CH1–CH50) and conjugated L band (CH51–CH100) channels.

**Table 1 sensors-23-02906-t001:** Configuration of optimized Raman amplifier.

	50 Transmitted Channels C Band 191.2–196.1 THz	50 Conjugated Channels L Band 186.2–191.1 THz
$P_{p}^{\pm }$ Wavelength	1370 nm	1410 nm
$P_{p_{2}}^{-}\;$ FBG Wavelength	1458 nm	1498 nm
$P_{p_{1}}^{+}$ Pump Power	1.3 W	0.7 W
$P_{p_{1}}^{-}$ Pump Power	1.838 W	1.717 W

## Data Availability

Original data is available at Aston Research Explorer. https://doi.org/10.17036/researchdata.aston.ac.uk.00000591.

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
