# Peer review of "Asymmetry Optimization for 10 THz OPC Transmission over the C + L Bands Using Distributed Raman Amplification"

_sensors, 2023, doi:10.3390/s23062906_

Round 1

Reviewer 1 Report

This paper is clearly presented. Theoretical and simulation results confirm the feasibility of its research program. The only shortcoming is the lack of experimental verification. Although the current simulation can basically accurately predict the experimental results, the simulation may not be accurate enough for signal amplification using wavelength conversion parametric amplification such as OPC. 

It is recommended that the author add a paragraph to discuss the shortcomings of his own simulation results and the actual system, or in which aspects the simulation can further improve the accuracy.

Reviewer 2 Report

In the attached file.

Reviewer 3 Report

Authors presented an optimized broadband Raman optical amplifier with C and L bands with improved performance.  Figure quality looks good. However, authors must emphasize the novelty in the manuscript and must revise the entire manuscript with enough time with suggestive comments as below.

1. Authors must edit and improve the manuscript with help of native colleague faculty or professional English service.

2. In titlte, please correct C + L band to C + L bands. 

3. In Line 11, We present an optimized design-> We presented optimized design

4. Authors need to show the novelty in Abstract section.

5. Authors mentioned [1-14] in Line 29. However, authors had better show their previous work briefly.

6. Authors need to show the research motivation and previous work in Introduction section.

7. In the preivous work, authors must describe the advantages and disadvantages of the previous work.

8. In the 1st equation, there is no reference citation.

9. In our search -> In our research in Line 90.

10. Authors need to change Fig. to Figure.

11. Please describe how to obtain the measured results in Figure 3. Please display the measurement setup.

12. In Figure 4, how to measure the measured asymmetry ?

13. What is unit of y-axis in Figure 5 ? % ?

14. In Figure 6, why two points were increased at 1460 nm ?

15. Authors had better show future work in Conclusion section.

Round 2

Reviewer 2 Report

Attached file.

Reviewer 3 Report

Authors corrected and answered all questions so I could recommend this article could be accepted as it is.

Author Response

We are glad to answer your questions and would like to thank You for your time on how to improve our manuscript.